# Imitation Learning using the Forward-Forward Algorithm

**Insik Chung, Isaac Han**
School of Integrated Technology
Gwangju Institute of Science and Technology
123 Chem-dan gwa-gi-ro, Gwangju 61005, Korea
{ischung1184, lssac7778}@gm.gist.ac.kr

**kyung-Joong Kim**
School of Integrated Technology
Gwangju Institute of Science and Technology
123 Chem-dan gwa-gi-ro, Gwangju 61005, Korea
{kjkim}@gist.ac.kr

## Abstract

The forward-forward (FF) algorithm has been recently introduced as a novel approach to training neural networks in a way that approximates the behavior of real neurons. Nevertheless, its application has been limited to visual domains and has not been investigated in the context of imitation learning or reinforcement learning. In this study, we evaluate the FF algorithm in the context of imitation learning. We implement a straightforward imitation model based on the FF algorithm and present a comparative analysis with the backpropagation (BP) model. Our findings indicate that the FF-based model exhibits comparable performance to the BP-based model on larger datasets but demonstrates inferior performance on smaller datasets.

## 1 Introduction

Reinforcement Learning (RL) has achieved significant success in diverse domains, including Atari (Mnih et al., 2015), Go (Silver et al., 2017), StarCraft (Vinyals et al., 2019), and protein folding (Jumper et al., 2021), leveraging the flexibility of deep learning in approximating policies and value functions. The stochastic gradient descent approach using backpropagation (BP) is widely employed for training neural networks in deep learning. However, recent research exploring the implementation of BP with real neurons (Lillicrap et al., 2016; 2020; Guerguiev et al., 2017) has revealed ambiguity in whether BP accurately corresponds to real neuron activities. To address this issue, Hinton (2022) proposes a forward-forward algorithm (FF) that replaces the forward and backward passes in BP with two forward passes.

Given this perspective, we consider imitation learning (IL), a problem of learning policies from static datasets, to be an intriguing problem. While FF has shown impressive results in the visual domain, it has not been studied in the context of imitation learning or reinforcement learning.

In this study, we evaluate the efficacy of the FF algorithm in the context of imitation learning and conduct a comparative analysis with the backpropagation (BP) model. We conducted experiments by changing the size of the training dataset in two classic environments. As a result, the FF imitation model performed similarly to the BP model on large datasets but performed worse than BP on small datasets.

## 2 Forward-Forward algorlithm

The Forward-Forward (FF) algorithm is a neural network learning algorithm that more closely resembles the learning structure of the human brain than the backpropagation algorithm. The FF algorithm replaces backpropagation's forward and backward passes with two forward passes. These two forward passes work with both positive (i.e., actual) and negative (i.e., generated) data. The network is trained to have high goodness in the positive data and low goodness in the negative data. Therefore, the network distinguishes positive data from negative data by comparing the goodness of the input data. Additionally, positive data include the correct label and negative data include the incorrect label, and the difference between positive and negative data is only the label. The network determines appropriate labels for input data.

## 3 METHOD

We considered imitation learning as a supervised problem that infers an action based on a given state by treating the state as an input and the action as a label. Therefore, by applying FF, the correct action was inferred from the given state in the same way as the positive data with the correct label was classified from the negative data.

Specifically, we used pairs of the correct action (considered as a label) and state as positive data, and pairs of incorrect action and state as negative data. To generate positive and negative data, the correct action for positive data was determined as the action most likely predicted by an expert-level model in a given state. Conversely, for negative data, an incorrect action was randomly selected from the action set, excluding the correct action. The action was then one-hot encoded and combined with the state to be utilized as positive or negative data.

For training, our model uses a metric called goodness, which is calculated as the sum of the squared outputs in every layer. The model has been trained to produce high goodness values for positive data and low goodness values for negative data. In the inference process, our model computes the goodness for all possible pairs of actions and states in a given state. It then selects the action associated with the highest goodness value. Through this process, our model classifies optimal and non-optimal actions based on the computed goodness.

## 4 EXPERIMENTS AND DISCUSSION

Our experiments aim to compare the forward-forward (FF) and the BP (BP) algorithms from the perspective of imitation learning. Thus, both models were built using a straightforward multi-layer perception model (MLP). We compared models on two standard environments, 'Cartpole' and 'MountainCar' (Brockman et al., 2016). The dataset for training was collected from an expert-level Proximal policy optimization (PPO) (Schulman et al., 2017) model. The dataset is a total of 100,000 frames, and each data consists of experience tuples of (state, action).

| Task | CartPole | | MountainCar | | |
|---|---|---|---|---|---|
| Data ratio | 1% | 5% | 10% | 50% | 100% |
| FF | 422.3 $\pm$152.1 | 500.0 $\pm$0.0 | -121.9 $\pm$27.5 | -112.9 $\pm$21.3 | -81.2 $\pm$15.2 |
| BP | 500.0 $\pm$0.0 | 500.0 $\pm$0.0 | -106.6 $\pm$10.6 | -104.6 $\pm$11.3 | -80.4 $\pm$14.0 |
| PPO(expert) | 500 $\pm$0.0 | | -80.0 $\pm$13.0 | | |

Table 1: Result of performance of Forward-Forward (FF) and Backpropagation (BP). Each value represents the average and standard deviation of the reward. PPO is an expert-level model that generated a dataset used to train FF and BP.

The performance of the FFIL model and the BPIL with various data sizes is compared in the table 1. In the 'CartPole' environment, FFIL and BPIL achieve 500 (maximum) rewards. However, FFIL requires more data than BP to reach the maximum reward. In the 'MountainCar', the experimental results indicate that FFIL, BPIL, and PPO (expert-level model) have comparable performance when trained on the entire dataset. However, as the size of the training dataset decreases, the performance of FFIL shows a considerable decline compared to BPIL.

Our analysis shows that FFIL is more sensitive to dataset size than BPIL. However, the FF paper does not provide insight into the relation between dataset size and performance. Consequently, we cannot ascertain whether this phenomenon is unique to the FFIL or a general characteristic feature of the FF model. One positive aspect is that FFIL can achieve a competitive level of performance with BPIL, given enough data. However, our approach is currently limited to discrete actions. Overcoming this limitation might increase FFIL's applicability to more imitation learning domains.

## 5 CONCLUSION AND FUTURE WORK

According to our investigation, FFIL performance is more sensitive to the size of the dataset than BPIL. To our knowledge, this is the first study that employs FF in the context of RL. Thus, FF can

be applied to RL in a wide variety of research. The development of an online RL model via FF or the implementation of an FFIL model that can infer action in continuous space were our next goals.

## 6 ACKNOWLEDGEMENTS

This research was supported by the National Research Foundation of Korea (NRF) funded by the MSIT (2021R1A4A1030075).

### URM STATEMENT

The authors acknowledge that at least one key author of this work meets the URM criteria of ICLR 2023 Tiny Papers Track.

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
