# OpenReview forum: "IMITATION LEARNING USING THE FORWARD-FORWARD ALGORITHM"
_ICLR.cc/2023/TinyPapers — Submitted to Tiny Papers @ ICLR 2023_

### Official Review · Reviewer_cj88 · 2023-03-22

**Confidence:** 4

**Summary Of Contributions:**

This paper proposes extending the recently proposed Forward-Forward (FF) algorithm to imitation learning. Results on imitating an expert PPO agent in typical toy environments show that an FF model can match a back-propagation model with sufficient samples. Future work includes development of FF towards online RL and continuous action spaces.

**Rating:**

High Potential (HP): a submission which meets the reviewing criteria and has potential to make an impact on the field

**Strengths And Weaknesses:**

Strengths:
* All claims and conclusions are well-justified with experimental results.
* Relevant literature is cited appropriately.
* The follow-up work is clear and promising.
* The paper is easy to read and grammatically well-written, except for some typos.
* The submission adheres to formatting requirements and the ICLR code of conduct.



Weaknesses:
* Some crucial definitions and explanations are missing from the methods and results.
* There are some minor typos that could have been caught with quick proofreading.

**Suggested Changes:**

Some explanations and definitions are missing. What is the architecture of each model? You mention the FF and BP models are MLPs, but don't mention further details such as network depth, layer widths, or nonlinearities, nor the details of the PPO expert. How were the architectures and hyperparameters selected? Then, in Table 1, I infer that the row "Dataset" means "The proportion of the dataset used", but this should be explicitly described. How were these sampled? Is each average/standard deviation measured across multiple models, or across multiple tests of the same agent?

This paper could benefit from a quick proofreading. There are a few typos: for example, in the first paragraph of Section 4, "...and the *back-propagation* (BP) algorithms...", "Therefore, ~Thus,~ both models...", and "*T*he experience dataset...". Further, ensure consistency of past vs. present tense: for example, Section 3 switches back and forth between tenses multiple times.

---

### Author Response · Authors · 2023-05-31
**For archival**

We wish to opt-in for archival

---

### Meta-Review · Area_Chair_1Fs6 · 2023-04-05

**Recommendation:** Invite to archive
**Confidence:** 4

**Metareview:**

Strengths
- Clear, organized and discusses relevant literature
- Claims match experiments

Weaknesses
- Important definitions are missing from the methods
- Insufficient details are included for reproducibility of experimental results; experiments only on two data sets.
- The negative result for a new algorithm for a new application should be complemented/compared with its original application

**Summary:**

The paper essentially provides a negative result, that the recently proposed Forward Forward algorithm does not seem to yield improvement over the standard back propagation algorithm for learning policies (RL) from static data ("imitation learning"), and in particular needs more data. The experiments are consistent with the claims, the message is clear from the presentation, but there are concerns about reproducibility.

**Comments And Feedback To The Authors:**

More details on experiments to ensure the results are reproducible by a researcher reading the paper, and some effort in proofreading and incorporating reviewer suggestions would be good to do.

Changes for a revision:
- elaborate on model/algorithm, hyperparameter and data-related details, without which the results would be hard to reproduce
- a proofreading pass e.g. "Finally, The action ...", "Goodness" is used before defining, etc.

Recommended:
- experiments on more datasets to verify/strengthen the claims made
- potentially investigate if the need for more data by FF also happens for the original applications in the visual domain or is this RL-specific

**Reason For Not Giving A Higher Recommendation:**

Main issue is reproducibility of results, since the experiments have not been described in sufficient detail nor any code has been provided.

**Reason For Not Giving A Lower Recommendation:**

The writing makes reasonably clear and the paper is generally correct (up to verifiability of experiments).

---

### Decision · Program_Chairs · 2023-04-07

Invite to archive